# Deblurring Neural Radiance Fields with Event-driven Bundle Adjustment

Yunshan Qi
State Key Laboratory of Virtual
Reality Technology and Systems,
SCSE, Beihang University
Beijing, China
qi_yunshan@buaa.edu.cn

Lin Zhu*
School of Computer Science and
Technology, Beijing Institute of
Technology
Beijing, China
linzhu@bit.edu.cn

Yifan Zhao
State Key Laboratory of Virtual
Reality Technology and Systems,
SCSE, Beihang University
Beijing, China
zhaoyf@buaa.edu.cn

Nan Bao
State Key Laboratory of Virtual
Reality Technology and Systems,
SCSE, Beihang University
Beijing, China
nbao@buaa.edu.cn

Jia Li*
State Key Laboratory of Virtual
Reality Technology and Systems,
SCSE, Beihang University
Beijing, China
jiali@buaa.edu.cn

## Abstract

Neural Radiance Fields (NeRF) achieves impressive 3D representation learning and novel view synthesis results with high-quality multi-view images as input. However, motion blur in images often occurs in low-light and high-speed motion scenes, which significantly degrades the reconstruction quality of NeRF. Previous deblurring NeRF methods struggle to estimate pose and lighting changes during the exposure time, making them unable to accurately model the motion blur. The bio-inspired event camera measuring intensity changes with high temporal resolution makes up this information deficiency. In this paper, we propose Event-driven Bundle Adjustment for Deblurring Neural Radiance Fields (EBAD-NeRF) to jointly optimize the learnable poses and NeRF parameters by leveraging the hybrid event-RGB data. An intensity-change-metric event loss and a photo-metric blur loss are introduced to strengthen the explicit modeling of camera motion blur. Experiments on both synthetic and real-captured data demonstrate that EBAD-NeRF can obtain accurate camera trajectory during the exposure time and learn a sharper 3D representations compared to prior works.

## CCS Concepts

• **Computing methodologies** → **Computer vision**.

## Keywords

Neural radiance fields, Event camera, Image deblurring, Novel view synthesis

---

*Correspondence should be addressed to Jia Li and Lin Zhu.
Website: https://cvteam.buaa.edu.cn

---

**ACM Reference Format:**
Yunshan Qi, Lin Zhu, Yifan Zhao, Nan Bao, and Jia Li. 2024. Deblurring Neural Radiance Fields with Event-driven Bundle Adjustment. In *Proceedings of the 32nd ACM International Conference on Multimedia (MM '24), October 28-November 1, 2024, Melbourne, VIC, Australia.* ACM, New York, NY, USA, 10 pages. https://doi.org/10.1145/3664647.3680569

## 1 Introduction

Neural Radiance Fields (NeRF) [27] achieves 3D implicit representation learning and photo-realistic novel view synthesis results with high-quality 2D images and precise camera poses as input. In low-light scenes, a camera often requires a longer exposure time to obtain an image with sufficient brightness [48], and a handheld camera may cause motion blur in the captured image. Moreover, a high-speed moving camera can also cause motion blur, even in bright scenes with short exposure time. The blurred images will cause NeRF to learn a blurry 3D implicit representation, resulting in degraded quality of the synthesized novel view images. Thus, it is a practical problem to reconstruct a sharp NeRF from blurry images when facing low-light and high-speed scenes [46].

For static scenes, motion blur is caused by the camera pose change during the exposure time. The core problem of reconstructing a sharp NeRF is accurately restoring the motion trajectory to better model the motion blur formation. Correspondingly, we need to establish a connection between scene radiance fields and camera pose change. Recent deblurring NeRF works try to resolve this by using a learnable blur kernel [20, 21, 25] or interpolating the camera poses with learnable poses at the start and end of the exposure [40]. However, they solely implicitly model the motion and are supervised by RGB images, which is insufficient to establish a strong connection between camera pose change and NeRF parameters. This results in unstable network training and a decrease in performance for complex and severe motion blur. As in Figure 1, Deblur-NeRF [25] and Bad-NeRF [40] all learn an inaccurate motion trajectory and an inferior NeRF with blurry rendering results.

The bio-inspired event camera can measure brightness change asynchronously with high temporal resolution [9], which makes up for the information loss in blurry images. Recently, $E^2NeRF$

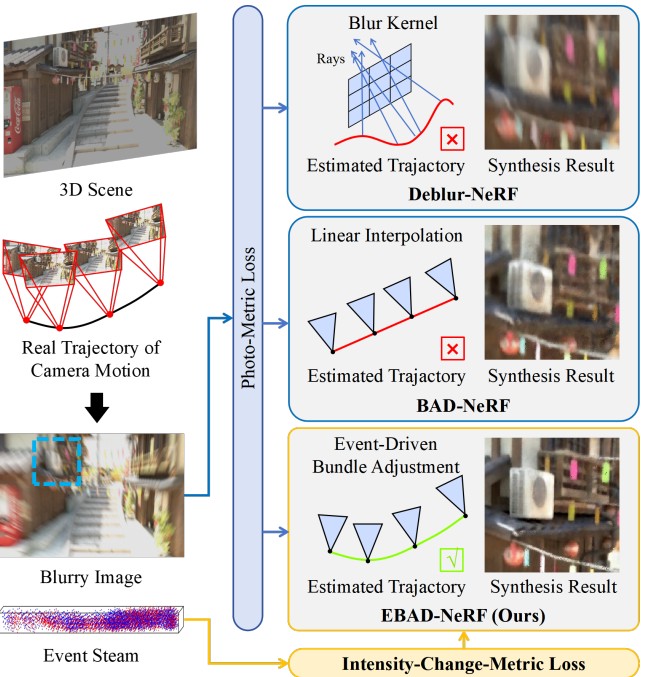

**Figure 1: Motivation of our proposed method. In a low-light scene or with a high-speed moving camera, motion blur usually occurs in the captured images. To reconstruct sharp NeRF from blurry images, Deblur-NeRF [25] uses a deformable sparse kernel to model the blur process. BAD-NeRF [40] linearly interpolates the camera poses and jointly learns the start and end poses of camera motion. However, these methods are unable to model complex motion blur and lack supervision information during exposure time with only a photo-metric loss. With the proposed event-driven bundle adjustment, our EBAD-NeRF can leverage the event to recover the accurate camera trajectory and learn a sharper NeRF, resulting in sharper novel view image rendering than the previous methods.**

[30] uses the event data and blurry images to enhance the model of motion blur in terms of image formation and achieves better results than the image-based deblurring works. However, it uses pre-deblurred images and COLMAP [32] to estimate camera poses, which is not learnable during training, and the pose change information in the event data is not fully modeled and utilized.

In this paper, we model the motion blur in terms of both motion trajectory and camera imaging. Event data is introduced to establish an explicit connection between the actual camera motion trajectory and the target sharp NeRF by conducting an intensity-change-metric event loss. Specifically, for each view, we first use multiple learnable camera poses to model the camera motion trajectory and transfer them into the **SE(3)** space [4, 40]. During the training, we jointly optimize these poses and the parameters of the NeRF network. We introduce an intensity-change-metric event loss to supervise pose changes during the exposure time. A photo-metric blur loss is also used to supervise the modeling of blurry

image formation. The image formation is also additionally strengthened by the event loss. We extend 5 blender synthetic scenes in Deblur-NeRF with event data and use a DAVIS-346 event camera [38] to capture spatial-temporal aligned Event-RGB (ERGB) real data with ground truth. Experiments on both synthetic and real data demonstrate that our method can learn superior NeRF and more accurate camera motion trajectories than previous image-based or ERGB-based deblurring NeRF methods. The rendering results are also better than the state-of-the-art image-based or ERGB-based image deblurring methods. To summarize, we present the following contributions:

1) A novel event-driven bundle adjustment deblurring neural radiance fields (EBAD-NeRF) framework is proposed to explicitly model image blur and jointly optimize the estimated camera motion trajectory and NeRF parameters.

2) An intensity-change-metric event loss and a photo-metric blur loss are presented to supervise the modeled motion blur in terms of both camera pose change and image formation conjugatively.

3) Experiments on extended synthetic data and real-captured data validate that our method achieves accurate motion trajectory estimation and high-quality 3D implicit reconstruction of the scene with severe blurry images and corresponding event data.

## 2 Related Work

### 2.1 Neural Radiance Fields

The neural radiation fields model achieves impressive novel view synthesis results and inspires a lot of subsequent research [10]. Some works [2, 3, 39, 45] improve the quality of learned geometry and synthesized novel views. Other works try to improve the training and rendering speed [8, 11, 15]. Besides, BARF [22], L2G-NeRF, and Nope-NeRF [5] explore reconstructing NeRF without camera poses estimated by COLMAP [32] and input images.

Blurry Images are often captured in low-light scenes or with a high-speed moving camera on drones or robots. Reconstructing sharp NeRF from blurry images also becomes a challenging problem. As shown in Table 1, Deblur-NeRF [25] proposes a deformable sparse kernel module to effectively model the blurring process. However, the blurring kernel is optimized based on 2D-pixel location independently. DP-NeRF [21] proposes a rigid blurring kernel and generates a 3D deformation fields, which is constructed as the 3D rigid motion of the camera for each view. Sharp NeRF [20] proposed a learnable grid-based kernel to obtain sharp output from neural radiance fields. Inspired by BARF, BAD-NeRF [40] conducts linear interpolation for the camera poses during the exposure time and jointly optimizes the poses with bundle adjustment.

### 2.2 Image Deblurring

Traditional deblurring algorithms focus on finding the suitable blur kernel of a blurry image to recover a sharp image. Hand-crafted features or sparse priors are used to tackle this problem [7, 19, 44]. Deep learning deblurring works directly learn an end-to-end mapping from blurry images to sharp images [36, 41, 51]. MPRNet [47] and SRN [37] achieve impressive single-image deblurring results. However, the information missing from blurry images during exposure time decreases the robustness of the image-based methods and

**Table 1: Comparison of Previous Deblurring NeRF Works**

| Method | RGB | Event | Motion Blur Trajectory Modeling |
|---|---|---|---|
| Deblur-NeRF | ✓ | - | 2D Blur kernel |
| DP-NeRF | ✓ | - | 3D Blur kernel |
| Sharp-NeRF | ✓ | - | Grid-based Blur kernel |
| BAD-NeRF | ✓ | - | Linear Interpolation Bundle Adjustment |
| $E^2$NeRF | ✓ | ✓ | Fixed Poses from Pre-deblurred Images |
| EBAD-NeRF | ✓ | ✓ | **Event-driven Bundle Adjustment** |

limits their performance when facing various and severe motion blur in low-light or high-speed scenes.

## 2.3 Event Camera

The event camera is based on a bio-inspired vision sensor that measures brightness changing asynchronously [9]. The high temporal resolution and high dynamic visual data capturing paradigm makes it ideal for flow estimation [1, 12, 14, 28, 34], feature detection and tracking [43, 49, 50] and images deblurring [23, 29, 33, 42]. Pan *et al.* [29] propose a simple event-based double integral model (EDI) based on event generation to establish the connection between blurry images and events. Other works [18, 33, 35] use learning-based networks to recover sharp images with events.

Recently, some event-based NeRF works have emerged. Ev-NeRF [17] models the measurement of the event sensor to learn grayscale neural radiance fields derived from the event stream. EventNeRF [31] reconstructs color NeRF with a color event camera. *e*-NeRF [24] improves Ev-NeRF and EventNeRF for non-uniform camera motion. SpikeNeRF [53] derives a NeRF-based volumetric scene representation from spike camera data. DE-NeRF [26] uses RGB images and events to learn a deformable NeRF. $E^2$NeRF is the first to reconstruct a sharp NeRF with blurry images and corresponding event data and achieves state-of-the-art performance. However, it only explicitly models the blurring processing at the imaging aspect. For camera motion trajectory during the exposure time, $E^2$NeRF uses EDI [29] to pre-deblurred images and COLMAP to estimate the camera poses that can not be optimized during training. Ev-DeblurNeRF [6] extends from DP-NeRF [21] with a novel EDI loss and an eCRF network to simulate the generation of events more realistically and achieves a remarkable deblurring effect. It also uses extra continuous events between spare image views to enhance the NeRF learning, which differs from the setting of $E^2$NeRF and ours.

## 3 Method

We introduce EBAD-NeRF to simultaneously learn the poses during the camera motion blur process and sharp neural radiance fields with blurred images and the event stream within the corresponding exposure time. Events serve two key purposes within the framework: (1) optimizing camera trajectories within the exposure time, and (2) contributing to the physical formation of motion blur in the image by providing information on light intensity changes. Moreover, we employ a photo-metric loss to supervise blur formation in the RGB domain. Figure 2 is the overview of our approach.

## 3.1 Motion Blur Formation in Static 3D Scene

In a static 3D scene, camera motion blur is caused by the changes of camera pose $\mathbf{P}$ during the exposure time. A digital image $\mathcal{I}$ is obtained by measuring the integration of light intensity $I$ with respect to time $t$ on the image sensor:

$$\mathcal{I} = \int_{t_{start}}^{t_{end}} I(t)\mathrm{d}t, \tag{1}$$

where $t_{start}$ and $t_{end}$ are the start and end time of the exposure period. Since light intensity $I$ is only corresponding to the camera pose in the static scene, we can express $\mathbf{P}$ as a function of time $\mathbf{P} = p(t)$ and a blurry image $\mathcal{B}$ can be expressed as:

$$\mathcal{B} = \int_{t_{start}}^{t_{end}} I(P(t))\mathrm{d}t. \tag{2}$$

In NeRF [27], given a camera pose $\mathbf{P} = P(t)$, for each pixel of the imaging plane, we can obtain a ray $\mathbf{r}$ that emits from the optical center of the camera and passes through the pixel $\mathbf{x} = (x, y)$. With stratified sampling, we can divide the part of this ray starting with $l_{near}$ and ending with $l_{far}$ into N equal parts and randomly sample one point in each part. For each of the $N$ sampled points, we input its 3D coordination $\mathbf{o}$ in the world coordinate system and the 2D view direction $\mathbf{d}$ represented by the ray $\mathbf{r}$ into NeRF MLP $F_\theta$ with network parameters $\theta$:

$$(\mathbf{c}, \sigma) = F_\theta(\gamma_o(\mathbf{o}), \gamma_d(\mathbf{d})). \tag{3}$$

The outputs are color $\mathbf{c}$ and density $\sigma$ of the point, and $\gamma(\cdot)$ encodes the input to a higher $K + 1$ dimension:

$$\gamma_K(x) = \{\sin(2^k \pi x), \cos(2^k \pi x)\}_{k=0}^K. \tag{4}$$

Then, we can obtain $N$ colors $\{\mathbf{c}_i\}_{i=1}^N$ and densities $\{\sigma_i\}_{i=1}^N$ of the sampled points. By conducting volume rendering:

$$C(\mathbf{r}, \mathbf{x}) = \sum_{i=1}^N T_i(1 - \exp(-\sigma_i \delta_i))\mathbf{c}_i,$$

$$\text{where} \quad T(i) = \exp(-\sum_{j=1}^{i-1} \sigma_j \delta_j), \tag{5}$$

we can get the final color value $C(\mathbf{r}, \mathbf{x}) = C(\mathbf{P}, \mathbf{x})$ of the pixel $\mathbf{x}$ passed by the ray $\mathbf{r}$ corresponding to the pose $\mathbf{P}$. $\delta_i = l_{i+1} - l_i$ is the distance between adjacent sampled points, and $T_i$ is the transparency between $l_{near}$ and the sampled point.

If we assume that the learned NeRF MLP is sharp, we can model the image motion blur by discretizing Eq. (2) with $p$ poses $\{\mathbf{P}_i\}_{i=1}^p = \{P(t_i)\}_{i=1}^p$ sampled evenly over the exposure time $t_{start}$ to $t_{end}$:

$$\hat{\mathcal{B}}(\mathbf{x}) = \frac{1}{p} \sum_{i=1}^p C(P(t_i), \mathbf{x}), \mathbf{x} \in \mathcal{X}, \tag{6}$$

where $\mathcal{X}$ represents the pixels of the image sensor. Since we use virtual sharp image color $C(P(t_i), \mathbf{x})$ to replace the light intensity $I(P(t_i))$ of Eq. (2), we need to multiply each item by a weight of time. As the poses are temporally evenly sampled, we can directly use an average as $\frac{1}{p}$ to represent the weight as in Eq. (6).

At this point, we establish a connection between image motion blur, camera pose, and neural radiation fields.

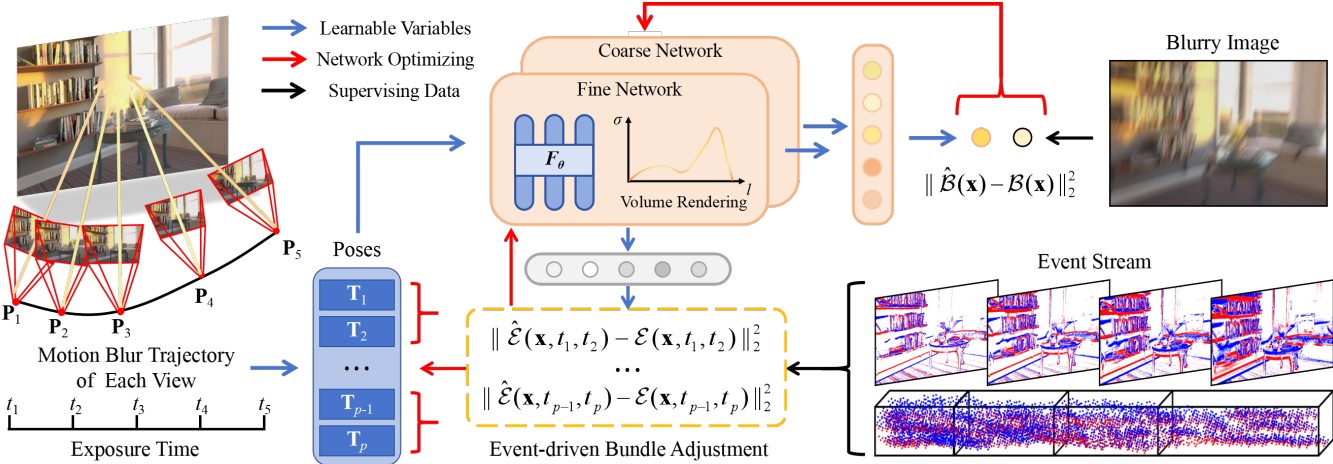

**Figure 2: Overview of our method. For each view of the static scene, an image with motion blur is caused by the camera moving during the exposure time. We temporal evenly sample $p$ poses $\{P_i\}_{i=1}^{p}$ on the motion trajectory and transfer them into $SE(3)$ as learnable variables $\{T_i\}_{i=1}^{p}$. With the poses, we model the blurry image and event generation in our network and jointly optimize the NeRF parameters $\theta$ and motion trajectory with the ERGB data to learn a sharp NeRF eventually.**

## 3.2 Event-driven Bundle Adjustment

In long exposure or high-speed situations, the camera is likely to have complex motion (non-uniform motion) during exposure time. To better model the motion blur, we transform the sampled $p$ poses $\{P_i\}_{i=1}^{p}$ into $p$ learnable pose $\{T_i\}_{i=1}^{p} \in SE(3)$ at time $\{t_i\}_{i=1}^{p}$. Then, we can use events as a bridge between camera pose changes and neural radiance fields to bundle these poses into actual camera motion trajectory during the NeRF optimization.

As described in Sec.3.1, we can express the virtual sharp image color during the exposure as $C(T_i, x) = C(P_i, x)$ and convert it into grayscale as $G(T_i, x)$ by averaging the RGB channels. Then, according to previous event simulation work [16], we transform $G(T_i, x)$ into the log function domain as $Ł(T_i, x)$ and simulate the generation of events caused by camera motion as:

$$\hat{\mathcal{E}}(x, t_i, t_{i+1}) = \begin{cases} \dfrac{Ł(T_{i+1}, x) - Ł(T_i, x)}{\Theta}, Ł(T_{i+1}, x) \leq Ł(T_i, x) \\ \dfrac{Ł(T_{i+1}, x) - Ł(T_i, x)}{\Theta}, Ł(T_{i+1}, x) > Ł(T_i, x) \end{cases}. \quad (7)$$

Then with the real events $\mathcal{E}(x, t_i, t_{i+1})$ captured by the event camera as event-driven bundle adjustment, we can optimize adjunctive poses $T_i$ and $T_{i+1}$ with the intensity-change-metric event loss by minimizing the differences between the numbers of predicted and actual events on all $m$ pixels $x \in X$ among all input views:

$$\mathcal{L}_{event} = \frac{1}{m}\sum_{x \in X} \frac{1}{p-1}\sum_{i=1}^{p-1} \|(\hat{\mathcal{E}}(x, t_i, t_{i+1})) - (\mathcal{E}(x, t_i, t_{i+1}))\|_2^2. \quad (8)$$

The computation of $\hat{\mathcal{E}}(x, t_1, t_2)$ is differentiable with respect to $\{T_i\}_{i=1}^{p}$ and latent sharp NeRF parameters $\theta$, which is optimized by the Jacobians of event loss:

$$\frac{\partial \mathcal{L}_{event}}{\partial \theta} = \frac{\partial \mathcal{L}_{event}}{\partial \hat{\mathcal{E}}(x)} \cdot \frac{1}{m}\sum_{x \in X} \frac{\partial \hat{\mathcal{E}}(x)}{\partial C(x)}\frac{\partial C(x)}{\partial \theta}, \quad (9)$$

$$\frac{\partial \mathcal{L}_{event}}{\partial T_i} = \frac{\partial \mathcal{L}_{event}}{\partial \hat{\mathcal{E}}(x)} \cdot \frac{1}{m}\sum_{x \in X} \frac{\partial \hat{\mathcal{E}}(x)}{\partial C(x)}\frac{\partial C(x)}{\partial T_i}, \quad (10)$$

Notice that in Eq. (8) we only calculate the event loss between adjacent poses because if the sampled poses are too far apart, the event loss will fluctuate greatly, reducing the training stability.

## 3.3 Final Loss

We conduct the photo-metric blur loss between the predicted and the input blurry images as in NeRF for all $m$ pixels $x \in X$ among all input views:

$$\mathcal{L}_{blur} = \frac{1}{m}\sum_{x \in X} \|\hat{\mathcal{B}}(x) - \mathcal{B}(x)\|_2^2, \quad (11)$$

The blur loss not only has a constraint on camera poses $\{T_i\}_{i=1}^{p}$ to avoid the spreading of poses but also plays a main role in supervising NeRF parameters $\theta$ to learn texture details of the scene that event data cannot provide:

$$\frac{\partial \mathcal{L}_{blur}}{\partial \theta} = \frac{\partial \mathcal{L}_{event}}{\partial \hat{\mathcal{B}}(x)} \cdot \frac{1}{m}\sum_{x \in X} \frac{\partial \hat{\mathcal{B}}(x)}{\partial C(x)}\frac{\partial C(x)}{\partial \theta}, \quad (12)$$

$$\frac{\partial \mathcal{L}_{blur}}{\partial T_i} = \frac{\partial \mathcal{L}_{blur}}{\partial \hat{\mathcal{B}}(x)} \cdot \frac{1}{m}\sum_{x \in X} \frac{\partial \hat{\mathcal{B}}(x)}{\partial C(x)}\frac{\partial C(x)}{\partial T_i}. \quad (13)$$

The final loss is defined as:

$$\mathcal{L}_{blur} = \lambda \mathcal{L}_{event}^{f} + \mathcal{L}_{blur}^{c} + \mathcal{L}_{blur}^{f}, \quad (14)$$

where $\lambda$ is a weight parameter of event loss. We used the design of the fine and coarse network in NeRF and conduct blur loss $\mathcal{L}_{blur}^{f}$ and $\mathcal{L}_{blur}^{c}$ on both of the networks. Event loss $\mathcal{L}_{event}^{f}$ is only calculated for the fine network because it can render enough texture details for precise event estimation. Then, the network can learn an accurate camera motion and latent sharp 3D implicit representation with gradient propagation.

## 3.4 Implementation Details

We train the EBAD-NeRF on a single NVIDIA RTX 3090 GPU. The training time is close to BAD-NeRF. The sample number of the rays is $N = 64$ for both fine and coarse networks. The number of sampled poses of each view $p$ is set to 5, and the weight of event loss $\lambda$ is set to 0.005. Sec.4.4 evaluates the influence of the parameters. Additionally, we use $\Theta = 0.3$ as the threshold of the event generation, which is a typical value in event-based vision [9]. A coarse initial pose for each view is given at the start of training.

## 4 Experiments

### 4.1 Datasets

*4.1.1 Synthetic Data:* We use the five Blender scenarios of Deblur-NeRF (Cozyroom, Factory, Pool, Tanabata, and Wine) to generate our synthetic data. We increase the camera shake amplitude to generate a severe blur and add the camera motion speed change during the exposure to simulate a non-uniform camera motion blur for the Factory, Pool, Tanabata, and Wine scenes. Besides, we input the virtual sharp frames during the camera shake process rendered by Blender into the V2E [16] to generate the corresponding event stream as in the synthetic datasets of $E^2$NeRF.

*4.1.2 Real Data:* To test the effectiveness of our method on real data, we capture two sets of real data (Bar and Classroom) with DAVIS-346 [38] event camera, which can capture spatial-temporal aligned event data and RGB data at the same time. Both sets of data are captured in low-light scenes, so RGB data requires a longer exposure time (100ms) to get a bright enough image. We use a handheld camera to capture blurry images and events for training and a tripod to capture sharp images for testing. Each scene contains 16 views of blurry images and corresponding events for training and 4 novel view sharp images for testing.

### 4.2 Comparing Methods and Metrics

*4.2.1 Image Deblurring Methods.* For comparison, we selected two classic learning-based image deblurring methods, MPR [47] and SRN [37]. In order to make a more fair comparison in terms of data modalities, we use D2Net [33], which also uses events and blurry image data to achieve image deblurring. We input the images deblurred by these three methods into the original NeRF for the novel view generation task and named them as MPR+NeRF, SRN+NeRF, and D2Net+NeRF.

*4.2.2 Deblurring NeRF Methods:* We selected image-based deblurring NeRF methods Deblur-NeRF [25] and BAD-NeRF [40] and ERGB deblurring NeRF method $E^2$NeRF [30]. All NeRF-based methods are trained with 5000 rays of batch size and 64 sampling points for coarse and fine networks. Other parameters are the default values of the methods. We perform 100,000 iterations on synthetic data. On real data, due to the reduction in image resolution and the number of input views, 50,000 iterations are enough for all NeRF-based methods to converge to the final results.

*4.2.3 Metrics:* We use PSNR, SSIM, and LPIPS [52] to evaluate the reconstruction results and use absolute trajectory error (ATE) to evaluate the fitting accuracy of the camera motion trajectory [13].

**Table 2: Quantitative Ablation Study on the Tanabata Scene.**

|  | PSNR↑ | SSIM↑ | LPIPS↓ | ATE↓ |
|---|---|---|---|---|
| BAD-NeRF | 20.65 | .7567 | .3349 | 0.0514 ± 0.0204 |
| EBAD-NeRF-noe | 22.00 | .7961 | .3728 | 0.0462 ± 0.0183 |
| EBAD-NeRF-linear | 23.75 | .8478 | .2881 | 0.0418 ± 0.0160 |
| EBAD-NeRF-full | **24.99** | **.8800** | **.2069** | **0.0301 ± 0.0122** |

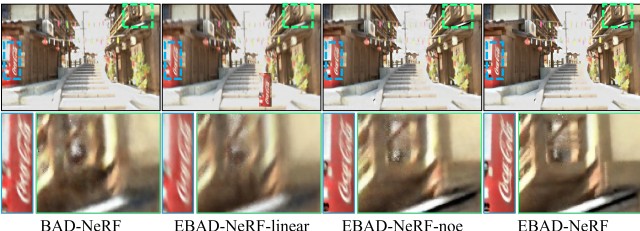

BAD-NeRF     EBAD-NeRF-linear     EBAD-NeRF-noe     EBAD-NeRF

**Figure 3: Qualitative ablation study on the Tanabata scene. EBAD-NeRF-linear and EBAD-NeRF-noe are defined in Sec. 4.3. With event-driven bundle adjustment, the rendering results of learned NeRF are sharper and clearer.**

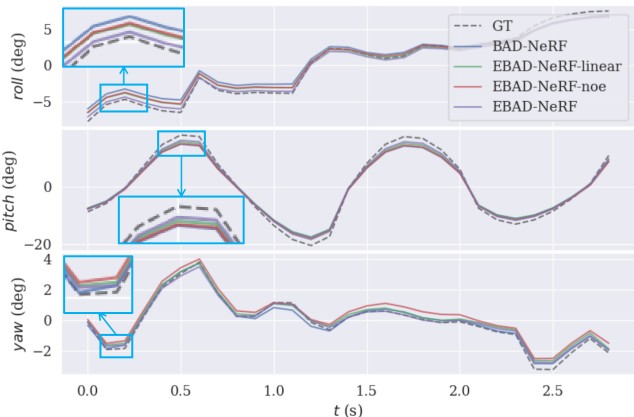

**Figure 4: Qualitative ablation study of trajectory fitting accuracy on Tanabata scene. EBAD-NeRF-linear and EBAD-NeRF-noe are defined in Sec. 4.3. EBAD-NeRF significantly fits the camera motion closer to the ground truth in the roll, pitch, and yaw metrics than all other methods.**

### 4.3 Event-driven Bundle Adjustment Analysis

We analyze the proposed event-driven bundle adjustment at two levels: 3D reconstruction quality and motion trajectory fitting accuracy. In Table 2 and Figures 3, 4, EBAD-NeRF-noe training the proposed EBAD-NeRF without event data as supervision and EBAD-NeRF-linear represents the results of adding event data constraints to BAD-NeRF with linear interpolation directly. Results in Table 2 are the average of the deblurring view and novel view.

*4.3.1 Defect of Linear Interpolation Bundle Adjustment:* The quantitative results in Table 2 demonstrate that without event enhancement, the bundle adjustment constrained by linear interpolation

(BAD-NeRF) is even worse than the unconstrained bundle adjustment (EBAD-NeRF-noe) when facing complex camera motion. With event enhancement, the same conclusion can be obtained by comparing EBAD-NeRF-linear and EBAD-NeRF-full.

*4.3.2   Effect of Event in Bundle Adjustment:* Comparing BAD-NeRF and EBAD-NeRF-linear, even with the negative impact of linear pose interpolation, EBAD-NeRF-linear still has improvements in trajectory fitting and reconstruction results with event enhancement as shown in Table 2. Comparing EBAD-NeRF-noe and EBAD-NeRF, introducing events significantly improves the accuracy of trajectory estimation and reconstruction results.

To sum up, during the NeRF training, event data can not only provide image-level optimization information but also pose-level practical constraints. Our proposed EBAD-NeRF effectively utilizes the ERGB-dual-modal data to achieve this goal. The qualitative comparison in Figure 3 and Figure 4 also testify to this, where EBAD-NeRF generates the sharpest rendering results and closest trajectory estimation to the ground truth.

## 4.4   Parameters Analysis

We evaluate the rendering results of the Cozyroom scene to analyze the influence of $p$ and $\lambda$ of our method in Figure 5.

*4.4.1   The influence of $p$:* Intuitively, the more virtual sharp frames, the more accurate the simulation of camera motion blur will be, and the corresponding network learning results will be better. The results in Figure 5 are consistent with this. When the number of virtual frames gradually increases from 1, the network's performance also significantly improves. However, after exceeding 5, the number of virtual frames has almost no impact on the reconstruction results, and more virtual sharp frames will require more calculations. Therefore, $p = 5$ is set in our experiments.

*4.4.2   The influence of $\lambda$:* As shown in Figure 5, when we take $\lambda$ as 0.005, EBAD-NeRF achieves the best results on the three metrics.

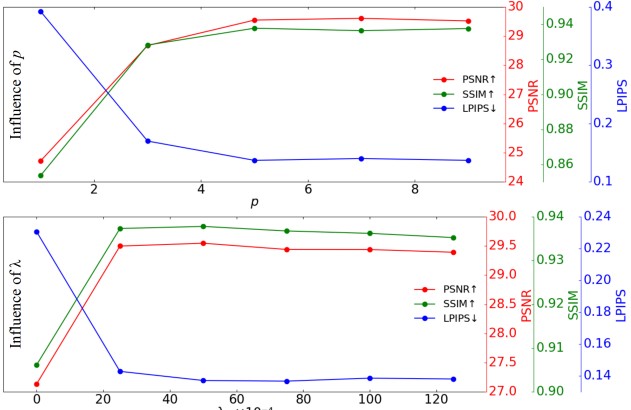

**Figure 5: Evaluation on the influence of the number of sampled poses $p$ and the weight parameter $\lambda$. The results are averages of reconstructed images on both deblurring and novel views of the Cozyroom scene. The red, green, and blue lines represent PSNR, SSIM, and LPIPS, respectively.**

**Table 3: Absolute Trajectory Error Results of Blender Scenes.**

|  | ATE↓ |
| --- | --- |
| COLMAP-Blur (NeRF [27]) | $0.0476 \pm 0.0171$ |
| COLMAP-EDI (E$^2$NeRF [30]) | $0.0419 \pm 0.0159$ |
| BAD-NeRF (Linear Interpolation[40]) | $0.0462 \pm 0.0142$ |
| BAD-NeRF (Cubic Interpolation[40]) | $0.0542 \pm 0.0207$ |
| Event-driven Bundle Adjustment (Ours) | $\mathbf{0.0383 \pm 0.0143}$ |

## 4.5   Quantitative Results

*4.5.1   Quantitative Results of Pose Estimation on Blender Scenes:* We use the Evo tool [13] to calculate the absolute trajectory error, considering both rotation and translation. The results in Table 3 are the average of five blender scenes, which shows the accuracy of camera motion trajectory estimation by different methods. COLMAP-Blur means directly estimating the camera poses with blurry images and COLMAP for each view. COLMAP-EDI represents the pose estimation method used in E$^2$NeRF, which uses the EDI [29] algorithm to first deblur the image with events and then input the sharp image sequence into COLMAP to estimate camera poses during the blur process. BAD-NeRF learns the start and end poses of camera motion. It uses linear interpolation to simulate the intermediate poses and extends it with a cubic interpolation. The results of these two methods are also shown in the table. Our proposed event-driven bundle adjustment method introduces event into joint learning of camera pose changes and sharp NeRF, achieving the best results in restoring the camera pose, which also promotes EBAD-NeRF to further improve the reconstruction quantification results as in Sec. 4.5.2 and Sec. 4.5.3.

*4.5.2   Quantitative Results of Reconstruction on Blender Scenes:* We evaluate the reconstruction results of five blender scenarios on both deblurring views (Table 4) and novel views (Table 5). SRN and SRN+NeRF achieve the best results among single-image-deblurring-based methods and even surpass the Deblur-NeRF. With the help of event data, D2Net and D2Net+NeRF are slightly better than MPR and MPR+NeRF. BAD-NeRF with cubic interpolation is further improved over BAD-NeRF with linear interpolation and SRN+NeRF, though it is still worse than event-enhanced methods E$^2$NeRF and our proposed EBAD-NeRF on both novel view and deblurring view.

Note that there is no apparent difference between the E$^2$NeRF and EBAD-NeRF on the Cozyroom scene where blur is only caused by linear camera motion, and E$^2$NeRF is even better than EBAD-NeRF on LPIPS. However, in the other four scenes with more complex camera shaking as described in Sec. 4.1.1, the results of EBAD-NeRF in both deblurring views and novel views are significantly better than E$^2$NeRF, which indicates that with fixed pre-estimated poses limits the effect of events in 3D implicit learning in E$^2$NeRF, especially when facing complex camera motion. Accordingly, EBAD-NeRF uses event-driven bundle adjustment to jointly optimize the motion poses of the camera, further releasing the potential of event data in reconstructing a sharp NeRF with blurry images.

*4.5.3   Quantitative Results of Reconstruction on Real Scenes:* To verify the effectiveness of our method on real data, we conducted experiments on real data mentioned in 4.1.2. As shown in Table 6,

**Table 4: Quantitative Results on Deblurring Views of Blender Scenes. The Best Results are Shown in Bold.**

| | Cozyroom | | | Factory | | | Pool | | | Tanabata | | | Wine | | | Average | | |
|---|---|---|---|---|---|---|---|---|---|---|---|---|---|---|---|---|---|---|
| | PSNR↑ | SSIM↑ | LPIPS↓ | PSNR↑ | SSIM↑ | LPIPS↓ | PSNR↑ | SSIM↑ | LPIPS↓ | PSNR↑ | SSIM↑ | LPIPS↓ | PSNR↑ | SSIM↑ | LPIPS↓ | PSNR↑ | SSIM↑ | LPIPS↓ |
| NeRF | 27.04 | .9034 | .2875 | 20.54 | .6776 | .5233 | 27.39 | .8352 | .4356 | 19.13 | .6691 | .5743 | 20.11 | .6732 | .5832 | 22.84 | .7517 | .4808 |
| D2Net | 28.25 | .9271 | .2083 | 21.15 | .7096 | .4428 | 28.23 | .8589 | .3537 | 19.34 | .6947 | .4973 | 20.60 | .6977 | .4894 | 23.52 | .7776 | .3983 |
| NeRF+D2Net | 28.11 | .9234 | .2256 | 21.20 | .7100 | .4485 | 28.23 | .8576 | .3757 | 19.48 | .6964 | .5077 | 20.64 | .6976 | .5097 | 23.53 | .7770 | .4134 |
| MPR | 26.08 | .9025 | .2505 | 21.07 | .6985 | .4503 | 27.09 | .8318 | .3692 | 19.42 | .7046 | .4983 | 20.31 | .6907 | .5172 | 22.80 | .7656 | .4171 |
| NeRF+MPR | 26.78 | .9094 | .2642 | 21.14 | .7023 | .4660 | 27.33 | .8388 | .4063 | 19.79 | .7131 | .5161 | 20.56 | .6936 | .5504 | 23.12 | .7714 | .4406 |
| SRN | 28.11 | .9216 | .1943 | 22.96 | .7752 | .3384 | 28.58 | .8705 | .2882 | 19.80 | .7202 | .4208 | 21.30 | .7297 | .4160 | 24.15 | .8034 | .3315 |
| NeRF+SRN | 28.29 | .9271 | .2073 | 23.22 | .7900 | .3309 | 28.88 | .8812 | .2964 | 20.11 | .7340 | .4238 | 21.61 | .7448 | .4215 | 24.42 | .8154 | .3360 |
| Deblur-NeRF | 28.49 | .9275 | .1993 | 21.67 | .7213 | .4431 | 27.99 | .8581 | .3527 | 18.99 | .6685 | .4843 | 20.65 | .6915 | .4914 | 23.56 | .7734 | .3942 |
| BAD-NeRF | 28.80 | .9278 | .1872 | 20.39 | .6696 | .4052 | 29.46 | .8867 | .2416 | 20.35 | .7475 | .3363 | 22.10 | .7540 | .3831 | 24.22 | .7971 | .3107 |
| BAD-NeRF-Cubic | 28.89 | .9314 | .1664 | 25.63 | .8526 | .2822 | 30.26 | .9049 | .2078 | 21.31 | .7677 | .3706 | 23.22 | .7923 | .3384 | 25.86 | .8498 | .2731 |
| E$^2$NeRF | 30.17 | .9459 | **.1057** | 27.90 | .9046 | .2638 | 29.29 | .8841 | .2420 | 24.02 | .8625 | .2906 | 25.63 | .8688 | .2919 | 27.40 | .8932 | .2388 |
| EBAD-NeRF | **30.53** | **.9475** | .1120 | **28.10** | **.9085** | **.1717** | **31.50** | **.9193** | **.1607** | **24.91** | **.8783** | **.2071** | **26.66** | **.8732** | **.2230** | **28.34** | **.9054** | **.1749** |

**Table 5: Quantitative Results on Novel Views of Blender Scenes. The Best Results are Shown in Bold**

| | Cozyroom | | | Factory | | | Pool | | | Tanabata | | | Wine | | | Average | | |
|---|---|---|---|---|---|---|---|---|---|---|---|---|---|---|---|---|---|---|
| | PSNR↑ | SSIM↑ | LPIPS↓ | PSNR↑ | SSIM↑ | LPIPS↓ | PSNR↑ | SSIM↑ | LPIPS↓ | PSNR↑ | SSIM↑ | LPIPS↓ | PSNR↑ | SSIM↑ | LPIPS↓ | PSNR↑ | SSIM↑ | LPIPS↓ |
| NeRF | 26.98 | .9021 | .2875 | 20.58 | .6953 | .5226 | 27.22 | .8345 | .4320 | 19.12 | .6882 | .5654 | 19.91 | .6763 | .5852 | 22.76 | .7593 | .4785 |
| NeRF+D2Net | 28.02 | .9219 | .2264 | 21.49 | .7345 | .4490 | 28.04 | .8582 | .3745 | 19.70 | .7181 | .4990 | 20.43 | .7137 | .5092 | 23.54 | .7893 | .4116 |
| NeRF+MPR | 26.60 | .9066 | .2665 | 22.08 | .7355 | .4629 | 27.20 | .8368 | .4045 | 19.81 | .7183 | .5112 | 20.45 | .6968 | .5513 | 23.23 | .7788 | .4393 |
| NeRF+SRN | 28.21 | .9251 | .2076 | 23.18 | .8018 | .3300 | 28.78 | .8808 | .2939 | 21.03 | .7708 | .4133 | 21.99 | .7637 | .4203 | 24.64 | .8284 | .3330 |
| Deblur-NeRF | 28.42 | .9260 | .2000 | 22.12 | .7490 | .4369 | 27.92 | .8594 | .3491 | 20.77 | .7412 | .4605 | 21.13 | .7128 | .4859 | 24.07 | .7977 | .3865 |
| BAD-NeRF | 28.52 | .9250 | .1918 | 21.11 | .7169 | .4077 | 29.70 | .8919 | .2376 | 22.37 | .8102 | .3267 | 22.79 | .7791 | .3812 | 24.90 | .8246 | .3090 |
| BAD-NeRF-Cubic | 29.14 | .9337 | .1677 | 23.25 | .7758 | .3179 | 30.68 | .9134 | .2047 | 20.37 | .7432 | .4260 | 23.82 | .8185 | .3358 | 25.45 | .8369 | .2904 |
| E$^2$NeRF | 30.07 | .9469 | **.1062** | 27.78 | .9090 | .2654 | 29.34 | .8905 | .2374 | 24.25 | .8730 | .2882 | 25.70 | .8772 | .2910 | 27.43 | .8993 | .2376 |
| EBAD-NeRF | **30.60** | **.9476** | .1138 | **28.25** | **.9146** | **.1583** | **31.62** | **.9244** | **.1559** | **25.45** | **.8898** | **.2056** | **26.72** | **.8889** | **.1961** | **28.53** | **.9131** | **.1659** |

**Table 6: Quantitative Results on Real Scenes. The Results are the Average of the Scenes, and the Best Results are Shown in Bold.**

| | NeRF | NeRF+D2Net | NeRF+MPR | NeRF+SRN | Deblur-NeRF | BAD-NeRF | BAD-NeRF-Cubic | E$^2$NeRF | EBAD-NeRF |
|---|---|---|---|---|---|---|---|---|---|
| PSNR↑ | 24.87 | 26.31 | 24.35 | 25.91 | 25.59 | 28.10 | 27.03 | 27.78 | **29.60** |
| SSIM↑ | .8562 | .8829 | .8539 | .8706 | .8744 | .8839 | .8744 | .9062 | **.9175** |
| LPIPS↓ | .5005 | .4092 | .4963 | .4198 | .3885 | .3632 | .2978 | .2403 | **.2156** |

the comparison results with other methods are highly consistent with the conclusions obtained in Tables 4 and 5, which verifies the reliability of our method in practical applications.

## 4.6 Qualitative Results

*4.6.1 Qualitative Results of Reconstruction on Blender Scenes.* We evaluate the reconstruction results of the deblurring view on the Pool scene (Figure 6) and the novel view on the Wine and Factory scenes (first two rows in Figure 7). Since image-deblurring methods cannot synthesize novel view images, we do not show their results in Figure 7). The results of EBAD-NeRF are sharper than all other methods on both deblurring and novel views. For the green plants and wooden boxes in the Pool scene, some artifacts appear in the results of E$^2$NeRF, and our method effectively recovers the texture details. In the Wine and Factory scenes, though E$^2$NeRF realizes

noticeable results for the characters and letters, our method is closer to ground truth in color, clarity, and line thickness.

*4.6.2 Qualitative Results of Reconstruction on Real Scenes.* As in the third row of Figure 7, our method better recovers the white letters on the blackboard compared to E$^2$NeRF. BAD-NeRF is limited in reconstructing smooth areas (white areas on the podium). The results of other image-deblurring-NeRF methods and Deblur-NeRF are all limited by the severe blurry input, which is consist with the quantitative results in Table 6.

*4.6.3 Qualitative Results of Pose Estimation on Blender Scenes.* In Figure 8, we compare the estimated camera trajectory of different methods from the three dimensions of roll, pitch, and yaw. The purple curve representing EBAD-NeRF is the closest to the ground truth dashed line, which is consistent with the results in Table 3.

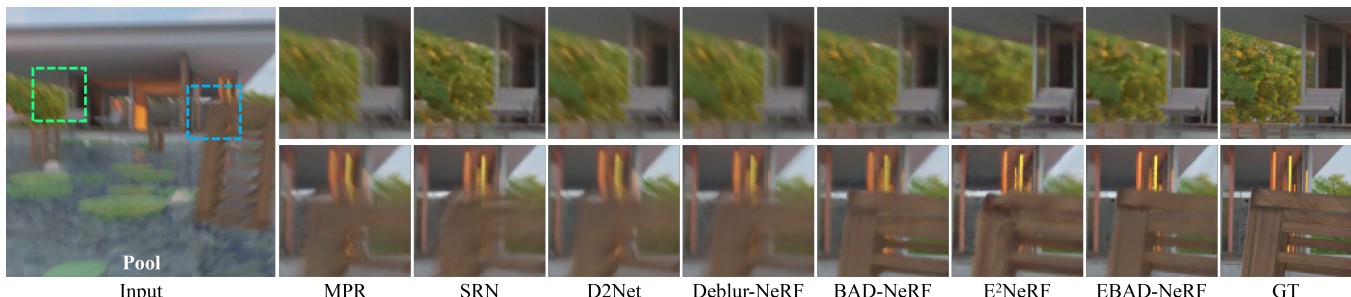

**Figure 6: Qualitative deblurring views rendering results of blender Pool scene.**

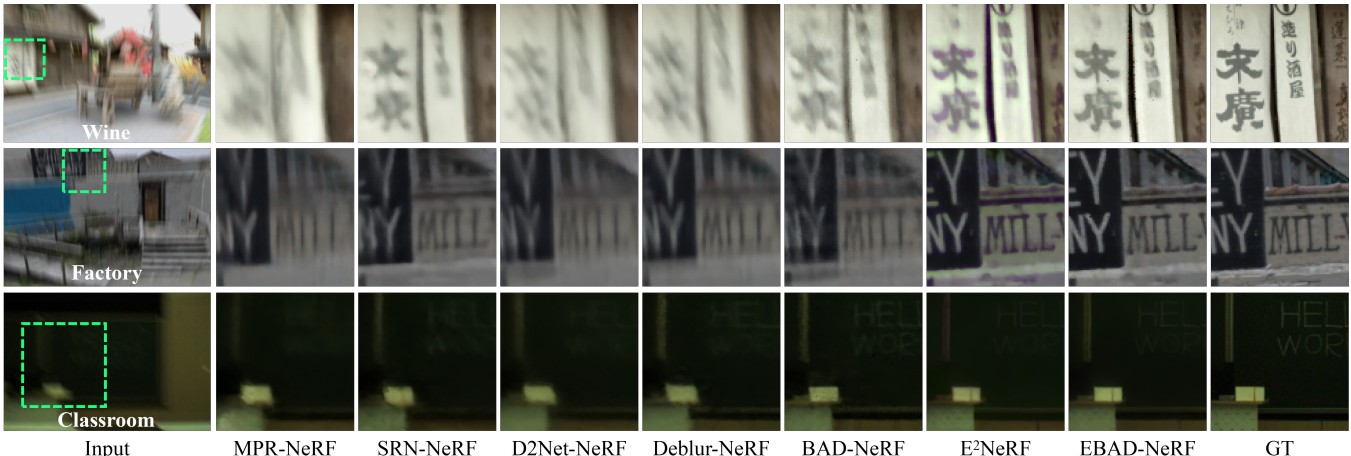

**Figure 7: Qualitative novel views rendering results of blender scenes (first and second row) and real scenes (third row).**

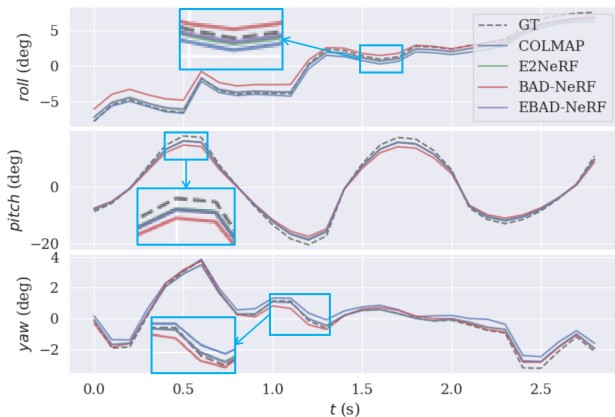

**Figure 8: Qualitative evaluation of trajectory estimation on Tanabata scene. EBAD-NeRF significantly fits the camera motion closer to the ground truth in the roll, pitch, and yaw metrics than all other methods. Although E$^2$NeRF uses predeblurred images with events, the estimated motion trajectory is still distorted, leading to reconstruction degradation even with event enhancement during training.**

Experiments on synthetic and real data both show that our method can learn better poses of the camera motion blur process with event-driven bundle adjustment compared to the previous deblurring NeRF methods. Upon this basis, the event data is also superimposed on the image blur process supervision, eventually achieving better 3D reconstruction and novel view synthesis effects.

## 5 Conclusion

In conclusion, this paper presents a novel approach, leveraging event-driven bundle adjustment, to address the challenge of modeling camera motion blur within neural radiance fields (NeRF). We draw inspiration from the emerging field of event-based vision, which offers high temporal resolution events, ideal for capturing dynamic information in motion blur. By integrating intensity-change-metric event loss and photo-metric blur loss, our framework enables the simultaneous optimization of blur modeling alongside NeRF reconstruction. Experiments on synthetic and real-captured datasets demonstrate the efficacy of our approach in accurately estimating camera poses and producing sharp NeRF reconstruction results.

## Acknowledgments

This work is partially supported by the National Natural Science Foundation of China under Grants 62132002, 62302041, and 62202010 and the China National Postdoctoral Program BX20230469.

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
