# OpenReview forum: "Deblurring Neural Radiance Fields with Event-driven Bundle Adjustment"
_acmmm.org/ACMMM/2024/Conference — MM2024 Poster_

### Official Review · Reviewer_QUX8 · 2024-05-24

**Rating:** 3
**Confidence:** 2

**Summary:**

This paper proposes Event-driven Bundle Adjustment for Deblurring Neural Radiance Fields (EBAD-NeRF) to strengthen the explicit modeling of camera motion blur.

**Strengths:**

A novel event-driven bundle adjustment deblurring neural radiance fields (EBAD-NeRF) framework is proposed to explicitly model image blur and jointly optimize the estimated camera motion trajectory and NeRF parameters

**Limitations:**

1.In 3.2,Event-driven Bundle Adjustment is not clearly stated.Formulas 7 and 8 only explain how events are triggered, but do not explain the representation of events in methods.There is no clear explanation of the advantages brought by introducing events.
2.In Table 2 and Table 3, Quantitative Results on the Tanabata Scene, The same scene got different results in PSNR SSIM etc.
3.In 4.1.2, The real data set is not introduced in detail, making the real data results in Table 5 and Figure 7 not confusing.
4.Suppose you compare your performance on the synthetic data set of the BAD-NeRF can better reflect the superiority of the improvement of your proposed method without events.

**Suitability:**

3

---

### Official Review · Reviewer_p8Qc · 2024-05-24

**Rating:** 4
**Confidence:** 3

**Summary:**

The authors consider a problem of bundle-adjusting NeRF with blur image input, where both camera poses and NeRF parameters should be jointly optimized. Additional event-RGB data is introduced to solve this problem and thus an intensity-change-metric event loss is proposed to model the camera motion blur during training. The authors carry out extensive experiments and compare with other baseline methods on various benchmark datasets, which validates the effectiveness of the proposed method.

**Strengths:**

Good writing. The manuscript is well-written overall. The motivation behind the authors' proposal is presented. All the steps in the proposed method are explained in a well-organized and easily understandable manner.
Extensive experiments. The authors performed comparisons with SOTA methods, which I think is sufficient to validate the effectiveness of the authors' method. The ablation study helps readers to better evaluate the contributions of this paper.
Impressive results. The authors' method outperforms the previous methods by notable margins both in synthetic and real-world datasets.

**Limitations:**

In my opinion, this is a nice paper to solve the joint optimization problem involving pose estimation and NeRF representation with blur image input. However, my concerns are as follows:
Lack of novel view synthesis results of event-camera. As event data is introduced to supervise this bundle-adjusting representation, the authors should provide quantitative results of rendered event images and show the comparison with the ground truth event data.
Lack of novelty. As far as I know, the authors just combine BAD-NeRF and E2NeRF to solve the joint optimization issue and further improve the performance of the baseline method, BAD-NeRF. Therefore, I give a borderline accpet. Please correct me if I am wrong and I will change my rating score.

**Suitability:**

3

---

### Official Review · Reviewer_A2ox · 2024-05-25

**Rating:** 2
**Confidence:** 4

**Summary:**

This paper introduces Event-driven Bundle Adjustment for Deblurring Neural Radiance Fields (EBAD-NeRF), which jointly optimizes learnable poses and NeRF parameters using hybrid event-RGB data. By incorporating an intensity-change-metric event loss and a photometric blur loss, EBAD-NeRF accurately models camera motion blur, resulting in improved camera poses and sharper 3D representations compared to previous methods.

**Strengths:**

1. Extensive results are conducted on private datasets.
2. Combing bundle adjustment with event information is a reasonable choice.

**Limitations:**

1. This work does not include a comparison with the most relevant research (Ev-DeblurNeRF, CVPR2024) in this area.
2. The method's novelty seems insufficient when compared to BAD-NeRF and E2NeRF. In my opinion, this work simply incorporates an event-guided loss (Eq.8) into BAD-NeRF, and this loss closely resembles the event loss employed in E2NeRF (Eq.9 in the E2NeRF paper).
3. Comparisons are only made using the proposed dataset. The effectiveness of this method could be further validated by comparing it to related works (such as E2NeRF, Ev-DeblurNeRF) on publicly available datasets like 1) https://github.com/iCVTEAM/E2NeRF, and 2) https://github.com/uzh-rpg/EvDeblurNeRF.
4. In Figure 1, BAD-NeRF produces poor results with linear interpolation. However, to my knowledge, BAD-NeRF is capable of performing cubic interpolation, which is available in its open-source code (https://github.com/WU-CVGL/BAD-NeRF). I am curious if the results would improve considering the expanded exposure time of the synthetic data used for this task.

As a result, I think this paper should not be accepted in its current state. I recommend authors to include the following results in the rebuttal:
1. Demonstrating the proposed method's superiority by comparing it with Ev-DeblurNeRF (CVPR2024) in synthetic and real datasets.
2. Offering key insights into the method's novelty (beyond simply combining bundle adjustment from BAD-NeRF with event loss).
3. Including additional comparison results on public datasets to further validate its effectiveness.

**Suitability:**

2

---

### Meta-Review · Area_Chair_Nfiw · 2024-07-02

**Recommendation:** Accept (Poster)
**Confidence:** 5

**Metareview:**

This paper was reviewed by three experts in the field. The recommendations are Borderline Reject, Borderline Reject, and Weak Reject. The main concerns raised by the reviewers include 1) Comparison to EvDeblurNeRF, 2) novelty, and 3) inapplicable to BAD-NeRF Dataset.

However, after serious consideration, ACs feel that these concerns are less critical. 1) EvDeblurNeRF is published after ACM MM submission deadline, and thus should be considered as concurrent work. 2) Although the idea of this work comes from BAD-NeRF and E2NeRF, integrating the event signal to BAD-NeRF is non-trivial. 3) BAD-NeRF Dataset does not include event data, and thus it is acceptable that the proposed solution does not work well on it.

Also, as all reviewers agree that this is the first work that uses event data for bundle adjustment, ACs feel this submission does bring enough contribution to the field. Therefore, the decision is to recommend the paper for acceptance to ACM Multimedia 2024.

Still, we recommended the authors to carefully read all reviewers’ final feedback and revise the paper accordingly. We congratulate the authors on the acceptance of their paper!